# The Fibrinolytic System and Its Measurement: History, Current Uses and Future Directions for Diagnosis and Treatment

**DOI:** 10.3390/ijms241814179

**Published:** 2023-09-16

**Authors:** Christine Lodberg Hvas, Julie Brogaard Larsen

**Affiliations:** 1Department of Anaesthesiology and Intensive Care, Aarhus University Hospital, 8200 Aarhus N, Denmark; c.hvas@clin.au.dk; 2Department of Clinical Medicine, Faculty of Health Sciences, Aarhus University, 8200 Aarhus N, Denmark; 3Department of Clinical Biochemistry, Regional Hospital Horsens, 8700 Horsens, Denmark

**Keywords:** fibrinolysis, plasminogen, plasmin, diagnostic tests

## Abstract

The fibrinolytic system is a key player in keeping the haemostatic balance, and changes in fibrinolytic capacity can lead to both bleeding-related and thrombosis-related disorders. Our knowledge of the fibrinolytic system has expanded immensely during the last 75 years. From the first successful use of thrombolysis in myocardial infarction in the 1960s, thrombolytic therapy is now widely implemented and has reformed treatment in vascular medicine, especially ischemic stroke, while antifibrinolytic agents are used routinely in the prevention and treatment of major bleeding worldwide. Despite this, this research field still holds unanswered questions. Accurate and timely laboratory diagnosis of disturbed fibrinolysis in the clinical setting remains a challenge. Furthermore, despite growing evidence that hypofibrinolysis plays a central role in, e.g., sepsis-related coagulopathy, coronary artery disease, and venous thromboembolism, there is currently no approved treatment of hypofibrinolysis in these settings. The present review provides an overview of the fibrinolytic system and history of its discovery; measurement methods; clinical relevance of the fibrinolytic system in diagnosis and treatment; and points to future directions for research.

## 1. Introduction

The fibrinolytic system, also termed the plasminogen–plasmin system, is an important physiological system and a key player in the haemostatic balance. Fibrinolysis is the process of fibrin degradation by plasmin and is activated when fibrin is formed as the end product of blood coagulation [1]. Thus, the fibrinolytic system is crucial in regulating intravascular fibrin deposition and clearance, maintaining haemostasis and facilitating wound healing while avoiding thrombosis. In addition, plasmin is involved in other important physiological processes such as tissue remodelling, angiogenesis, and inflammation [2,3].

The discovery and characterisation of the components of fibrinolysis as we know them today were made in the 1900s; thus, our knowledge of the fibrinolytic system is relatively new and is still expanding. A major breakthrough was the discovery of exogenous plasminogen activators in the 1930’s [4], which led to the development of thrombolytic agents, initially used for treatment of pleural adhesions and later for treating coronary artery thrombosis. Recombinant plasminogen activators were approved for use in the 1980’s and are still used worldwide for the treatment of myocardial infarction, pulmonary embolism, and ischemic stroke. Simultaneously, antifibrinolytic agents are increasingly used to prevent and treat major blood loss in trauma, postpartum bleeding, and surgery, and prehospital administration of antifibrinolytic drugs is now standard care for major trauma worldwide [5,6].

Over the years, the process of fibrinolysis has been subject to a multitude of studies in the laboratory, but in vivo fibrinolytic capacity can be challenging to quantify in an accurate and timely fashion, since fibrin clot degradation occurs slowly compared with other enzymatic processes due to the presence of endogenous inhibitors in plasma, and since total fibrinolytic capacity is determined not only by plasmin activity in plasma but also by fibrin clot structure and interactions with blood cells and endothelium. While a plethora of fibrinolysis assays have been developed and are in use for research [7], very few options currently exist for measuring fibrinolysis in the clinical setting. Viscoelastic tests may provide clinically relevant information on the patient’s fibrinolytic capacity, especially with modifications to existing standard protocols, but remain to be validated further in clinical settings.

This review presents a history of the discovery of fibrinolysis and our current understanding of the fibrinolytic system. It provides an overview of available laboratory methods to measure fibrinolysis, and of different clinical conditions in which disturbed fibrinolysis plays a role. Finally, the fibrinolytic system as a treatment target and future perspectives are discussed.

## 2. Historical Discovery of the Fibrinolytic System and Current Concepts

It was discovered by the Italian anatomist and physician Giovanni Battista Morgagni in the 18th century that in some cases, blood does not coagulate after sudden death. In 1861 Denis described spontaneous clot dissolution post mortem in trauma patients [1]. In the late 19th century, it was observed that dissolved fibrin could not be brought to clot again despite the addition of thrombin. This indicated that fibrin clot breakdown was the result of enzymatic digestion [8]; in 1889, Denys and De Marbaix postulated the existence of a dormant enzyme that could dissolve blood clots, and the term “fibrinolysis” appears to have been used for the first time by Dastre in 1893 [9].

The exact mechanisms behind fibrinolysis and the regulation of the process were still unknown, but the first half of the 20th century saw a rise in research on the fibrinolytic system, including the development and refinement of laboratory techniques. Gradually, the different components of the fibrinolytic system were elucidated. It became evident that a “dormant” enzyme (what we today call a zymogen) did indeed circulate in plasma and could be activated by different substances. The terms “plasminogen” and “plasmin” were proposed by Christensen and MacLeod in 1945 and were quickly adopted [10]. It was also recognised that plasma contained one or more inhibitors of fibrinolytic activity and that this inhibitory effect could be inactivated by substances such as chloroform or alleviated through plasma fractionation. A seminal paper by MacFarlane and Biggs in 1948 (75 years ago at the time of writing) summarised state-of-the-art knowledge of the fibrinolytic system at the time, establishing plasmin as the main fibrinolytic enzyme, present in zymogen form in the globulin fraction of plasma, and antiplasmin (now known as α2-antiplasmin) as the main inhibitor of fibrinolysis, present in the albumin fraction [9].

It was well known at this time that fibrin clots would dissolve under physiological conditions if left standing, albeit slowly, indicating spontaneous plasmin activity; but some conditions could induce a pronounced increase in plasmin activity, e.g., trauma, surgery, “shock”, and sudden death. It was considered that the release of a kinase (or plasminogen activator) into the bloodstream was involved, and the source was much sought after from the early 1900s and forward, with extensive investigations into the fibrinolytic activity of different tissues [9]. In 1952, Astrup and Stage succeeded in extracting the “fibrinokinin” from porcine heart tissue in water-soluble form [11]. It was discovered later that tissue plasminogen activator (tPA), as we know it today, is synthesized in the endothelium and released upon endothelial stimulation [12,13], and thus is present in tissue throughout the body. It was also recognised that fibrinolysis inhibitors other than antiplasmin existed and probably exerted their antifibrinolytic effects through plasminogen activator inhibition; however, it was not until the 1980s when specific assays and cloning techniques became available that plasminogen activator inhibitor-1 and -2 (PAI-1 and PAI-2) were characterised.

The interest in extrinsic plasminogen activators gained speed after the observation by Tillett and Garner in 1930s that streptococcal cultures contained a fibrinolysis activator, which appeared to work through the activation of plasminogen, later termed streptokinase [4]. The therapeutic potential was obvious, and in 1949, thrombolytic therapy with streptokinase was used for the first time for the treatment of pleural adhesions [14]. The following decades saw many studies on the use of streptokinase and later urokinase and tPA for the dissolution of intravascular thrombi in, e.g., myocardial infarction, pulmonary emboli, and peripheral artery disease. Though thrombolysis has been widely replaced by primary percutaneous coronary stenting (PCI) in the setting of myocardial infarction, thrombolysis with recombinant tPA is still used for the treatment of pulmonary embolus [15] and, most notably, in the treatment of ischaemic stroke where the beneficial effect on mortality and functional outcome after stroke has been firmly established [16].

Fibrinolytic activity in urine was described in 1947 [9], which led to the discovery of another plasminogen activator, initially named urokinase or urokinase-type plasminogen activator (uPA), present not only in urine but also in blood and extracellular matrix. It was discovered that uPA could induce cell migration and tissue degradation, and the cell-bound uPA receptor (uPAR) was described and characterised in the 1980s [17]. During inflammation, uPAR is shed from cell surfaces and circulates in soluble form (suPAR). Since its discovery at the beginning of the 1990s [18], the potential of suPAR as a diagnostic and prognostic biomarker has been explored in a range of clinical conditions, including cancer, cardiovascular disease, sepsis, and acute and chronic kidney disease [19,20,21]. Whether suPAR is merely a marker of immune activation or plays an active part in pathophysiological processes has been discussed; however, it has been documented that suPAR is a scavenger of circulating uPA and is a chemotaxant for neutrophils [22]. Thus, it could be a future treatment target in a range of inflammatory conditions.

As molecular studies of plasminogen and plasmin were carried out using newer techniques, the list of known substrates for plasmin expanded, and today we know that plasmin cleaves not only fibrin but also a wide range of proteins, including coagulation factors, hormones, growth factors and extracellular matrix proteins [3]. This has led to the realisation that the fibrinolytic system—or the plasminogen–plasmin system—is not only an important player in thrombosis and haemostasis, but is also involved in many other processes including tissue remodelling, angiogenesis, trophoblast invasion, neural development and inflammation [23,24,25,26,27]. These effects are crucial in physiological growth and development, but may also contribute to pathophysiological processes such as tumour growth and invasion, angioedema, or neovascularisation seen in, e.g., age-related macular degeneration [28,29,30]. These topics have been reviewed in detail by others [2,3] and are outside the scope of the review, but open interesting perspectives for future treatment of inflammatory and malignant diseases.

### Our Current Understanding of Fibrinolysis

After the activation of the coagulation system and thrombin generation, thrombin converts fibrinogen to fibrin through cleavage of the A and B domains (Figure 1). This allows the fibrin molecules to polymerise and form the insoluble fibrin clot, which is then further stabilised via covalent cross-linking by coagulation factor (F) XIIIa. The cross-linked fibrin exposes lysine residues, which provide a binding surface for plasminogen [31].

Intravascular conversion of plasminogen into plasmin is induced mainly by tPA. This serine protease is constitutively active but its plasminogen conversion activity increases up to 1000-fold in the presence of fibrin [32]. This ensures that under physiological conditions, intravascular plasmin activity is localised to fibrin clots and disseminated plasmin activation is avoided. In contrast, uPA activation requires binding to its receptor uPAR present on immune cells, endothelial cells, megakaryocytes, and in other tissues, but does not require co-localisation with fibrin [22]. This determines the localisation of uPA activity mainly to cell surfaces.

Fibrinolysis is regulated by several antifibrinolytic proteins, of which α2-antiplasmin is considered the most important plasmin inhibitor. A2-antiplasmin is incorporated into the fibrin clot during the fibrin cross-linking and inhibits plasmin directly via plasmin-antiplasmin (PAP) complex formation [33]. Other important regulators of fibrin clot breakdown are PAI-1, PAI-2, and thrombin-activatable fibrinolysis inhibitor (TAFI). The PAIs are serine protease inhibitors (SERPINs) and exert their antifibrinolytic effects through direct complex formation and the inhibition of tPA and uPA. PAI-1 is synthesised in the endothelium and in platelets and is the most abundant plasminogen activator inhibitor in vivo, while PAI-2 is synthesised mainly in placental tissue and may be involved in the regulation of trophoblast invasion [34]. The carboxypeptidase TAFI is synthesised in the liver and activated by thrombin in the presence of thrombomodulin. The activated protease (TAFIa) cleaves lysin residues from fibrin, and since the lysin residues are binding sites for plasminogen, TAFI effectively impairs plasminogen–fibrin binding, plasminogen activation, and fibrin degradation [35].

Not only plasma concentrations of pro- and antifibrinolytic proteins, but also the structure of the fibrin clot itself influence the fibrinolytic process. The properties of the fibrin network, i.e., fibre diameter, density, and pore size, determine clot stability and lysis resistance. These properties are highly dependent on concentrations of fibrinogen and thrombin at the time of clot formation. At low thrombin concentrations, the fibrin clot consists of thick fibres with a loose clot structure and large pore size highly susceptible to fibrinolysis. In contrast, higher thrombin concentrations yield thinner fibrin fibres and denser fibrin clots with smaller pores [36]. The denser fibrin structure hinders the diffusion of plasminogen and tPA and thereby renders the clot more lysis-resistant [37,38]. Furthermore, enhanced FXIII and TAFI activation by thrombin will also increase clot stabilisation by FXIIIa and decrease clot lysability through TAFIa lysin residue removal. The plasma concentration of fibrinogen also impacts fibrin clot properties; higher fibrinogen levels result in more compact and lysis-resistant clots [39]. Moreover, qualitative alterations of fibrinogen, such as phosphorylation, glycation, or oxidation can lead to altered clot structure and thereby possibly to impaired fibrinolysis [40,41].

## 3. Measuring Fibrinolysis

Fibrinolysis testing in a clinical setting has historically lagged far behind coagulation testing, in particular tests sensitive to decreased fibrinolytic capacity (hypofibrinolysis) [7]. This is for several reasons. First, the process of fibrinolysis is slow (hours–days) compared with coagulation (seconds–minutes); therefore, it is difficult to design tests with a clinically relevant turnaround time, which reflect in vivo fibrinolysis. Second, it has been difficult to show consistent associations between laboratory markers of fibrinolysis and clinical outcomes, possibly due to lack of intra- and inter-laboratory standardisation; thus, the definitions of clinically relevant hypo- and hyperfibrinolysis are not clear-cut. Furthermore, the therapeutic options for modulating fibrinolysis have been limited compared with options for anticoagulant medication and haemostatic agents, and thus the need to develop reliable assays for monitoring of fibrinolysis has been less pronounced. Nonetheless, a wide array of methods for measuring fibrinolysis is currently available in the research laboratory. The emergence of the turbidity clot formation and lysis assay, fluorescence assays for plasmin generation, development in microscope imaging methods, and, in recent years, modification of viscoelastic tests for increased sensitivity to hyper- and hypofibrinolysis, have all contributed to fibrinolysis research and have furthered our understanding of fibrinolysis. Below, an overview is given on the most widely used methods for measuring fibrinolysis.

### 3.1. Early Observations and Methods

The earliest observations of fibrinolytic activity were based on visual inspection of clot formation and subsequent dissolution in whole blood or plasma. This gave an excellent functional view of the patient’s intrinsic fibrinolysis speed, but did not provide information on underlying mechanisms and under physiological conditions, and it was a slow process (many hours to obtain spontaneous lysis of the clot). An important step was the development and refinement of fibrin plate techniques, which used preformed fibrin clots obtained by clotting purified fibrinogen with thrombin in a Petri dish under standardized conditions [42,43]. The sample of interest was then added onto the plate and the diameter of the lysed area was measured at specified time points. The fibrin plate technique allowed the study of fibrinolytic activity in different plasma fractions and tissue extracts, and thus facilitated much of early research on the fibrinolytic system.

### 3.2. Dynamic Assays

#### 3.2.1. Euglobulin Lysis Time and Plasma-Based Clot Formation and Lysis Assays

The euglobulin lysis time (ELT), based on early fibrin plate techniques, is still performed today and exists in several variant forms [44,45]. Common to the different methods is that the (eu)globulin fraction is separated from plasma for analysis to avoid the influence of α2-antiplasmin (coincidentally, PAI-1 and TAFI are also excluded). The globulin fraction can be added to a preformed fibrin clot, or the patient’s own fibrinogen, contained in the globulin fraction, can be clotted with thrombin. In both cases, lysis time is subsequently registered. In early days, the assay was performed using Petri dishes and visual inspection as described above, while nowadays microtiter plates are used and fibrin breakdown is assessed with automated continuous absorbance reading. The ELT is sensitive to the fibrinolytic capacity of the patient’s plasminogen/plasmin-tPA system, but it has the disadvantage of excluding the effect of antifibrinolytic proteins. Furthermore, the preformed fibrin clot techniques do not take the patient’s own fibrin structure and lysis resistance into consideration.

Clot formation and lysis assays performed on plasma (as opposed to the euglobulin fraction) solved some of these problems. In these assays, plasma is clotted using either thrombin or tissue factor, and fibrin formation and breakdown is assessed by measuring changes in plasma turbidity. However, as lysis is much slower in plasma than in the globulin fraction because of the presence of inhibitors, an exogenous plasminogen activator, e.g., recombinant tPA, is added simultaneously with the clot activator. Thus, these assays are not as sensitive to the patient’s endogenous tPA activity as the ELT, but provide a picture of the patient’s overall fibrinolytic capacity, including the effect of fibrinolysis inhibitors and, especially for variants of the assays using tissue factor instead of thrombin, also of the patient’s own clotting capacity and fibrin clot lysability. Clot formation and lysis assays are widely used for research purposes but are difficult to standardise between laboratories, as they are highly sensitive to reagents, buffers and even equipment [46].

#### 3.2.2. Plasmin Generation

Assessing the kinetics of plasmin formation and inhibition in plasma is another way to look at the patient’s dynamic fibrinolytic capacity. The plasmin generation assay was inspired by the similar thrombin generation assay and exists in several variants [47]. The assays use a fluorogenic substrate with high specificity for plasmin and continuously measure fluorescence after sample activation with tissue factor and tPA. The fluorescent signal is converted to plasmin activity units and different variables can be derived from the plasmin generation curve, e.g., time to plasmin formation, velocity, peak plasmin activity, area under curve, etc. Importantly, the assay not only gives information on plasmin formation, but also on the rate of plasmin inhibition, i.e., it is sensitive to α2-antiplasmin and other plasmin inhibitors. However, like the clot formation and lysis assay, exogenous tPA is added to enhance fibrinolysis and so the assay is less sensitive to the effect of the patient’s endogenous tPA.

#### 3.2.3. Fibrin Clot Structure

Besides plasmin activity, fibrin clot structure, i.e., fibre thickness, fibrin density and pore size, are the main determinants of lysis time. It can be assessed in the microscope or determined through measuring fibrin clot permeability. As microscope-imaging techniques have improved and become more widely available over time, this method is valuable for research into determinants of fibrin clot structure and it can be applied on clots generated from both plasma and whole blood [41]. The fibrin clot permeability method is performed by clotting the patient’s plasma with thrombin and letting buffer flow through the formed fibrin clots under standardised pressure. The permeability constant is calculated and is proportional to pore size and inversely proportional to fibrin clot density. This method is widely applied for research purposes [48].

#### 3.2.4. Viscoelastic Tests

Viscoelastic tests include thromboelastography (TEG^®^), rotational thromboelastometry (ROTEM^®^), and Sonoclot^®^. Tests are performed in whole blood and the kinetics of clot formation, as well as clot breakdown, is recorded through measuring the changing viscoelastic properties of the clotting blood, either as mechanic resistance between a rotating pin-and-cup system or as ultrasound waves. The viscoelastic tests are implemented worldwide to guide transfusion strategy in the bleeding patient [49]. Clot breakdown in the cup will change the viscoelastic properties, and therefore these analyses are sensitive to pronounced hyperfibrinolysis and are used to guide treatment with antifibrinolytic agents [50]. However, milder cases of hyperfibrinolysis will not be registered [51]. Furthermore, standard viscoelastic protocols do not contain fibrinolysis activator and, since physiological fibrinolysis is slow, they will not reveal decreased fibrinolytic activity within standard runtimes (60 min). This is illustrated by the fact that 0% lysis is contained in the reference intervals of standard protocols. Modified protocols with added tPA or urokinase have recently been developed for research use with both ROTEM^®^ and TEG^®^ [52,53,54] and one is available with Sonoclot^®^, and they have the potential to assess fibrinolytic capacity in whole blood with clinically relevant runtimes. However, data on association with clinical outcomes are still awaited.

### 3.3. Measurement of Circulating Factors

#### 3.3.1. Circulating Pro- and Antifibrinolytic Proteins

Specific assays are available either in-house or commercially for most of the pro- and antifibrinolytic proteins known today, including plasminogen, α2-antiplasmin, PAP complex, tPA, PAI-1 and -2, and TAFI. These are typically either chromogenic activity or antigen assays employing immunometric methods. They are widely used in research, but the clinical relevance of measuring single protein markers to assess fibrinolysis is currently limited, except perhaps in the face of suspected inherited deficiencies of, e.g., PAI-1 (which may lead to increased bleeding tendency) [55] or plasminogen (which is not associated with increased thrombosis risk but with ligneous conjunctivitis) [56]. Furthermore, there is lack of standardisation between assays, and for tPA and PAIs the additional problem exists that the available assays will have varying degrees of specificity for free protein versus tPA-PAI complex or active versus latent PAI-1 [57]. This can make interpretation tricky.

#### 3.3.2. Fibrin Degradation Products

Measurement of fibrin degradation products (FDPs) in plasma has been performed for decades [58], and FDPs in some form are among the most commonly investigated coagulation biomarkers worldwide. Semiquantitative assays and radioimmunoassays for FDPs were introduced in the early 1970s [59,60]. However, there were problems with standardisation as FDP’s are heterogeneous, and cross-reactivity with fibrinogen and fibrinogen breakdown products could not be avoided. The fibrin D-dimer fragment was discovered and isolated in the 1970s [61]. Fibrin D domain crosslinking only takes place after fibrin formation and FXIIIa crosslinking, and the fibrin D-dimer fragment is only released from the clot upon plasmin digestion; thus, the presence of fibrin D-dimer in plasma signifies ongoing fibrin formation and breakdown. Furthermore, the fibrin D-dimer is more well-defined and homogeneous than the broad array of FDPs previously investigated, and this facilitated the development of D-dimer-specific antibodies with much less potential for cross-reaction. The first ELISAs were developed in the late 1980s, and fibrin D-dimer measurement was in clinical use in 1991 [62]. This marker has since been implemented in the diagnosis of various prothrombotic conditions, most notably venous thromboembolism and disseminated intravascular coagulation. However, as fibrin D-dimer and other FDPs reflect both fibrin formation and breakdown, they are not particularly specific for fibrinolysis speed, and plasma levels of FDPs have not been demonstrated unambiguously to be good markers for increased or decreased fibrinolytic capacity. Other routine coagulation assays, such the prothrombin time (PT/INR) and activated partial thromboplastin time (aPTT), reflect clotting times only and are not sensitive to fibrinolytic capacity at all. Thus, we still lack reliable markers for fibrinolysis in the routine coagulation laboratory.

## 4. Fibrinolysis in Specific Clinical Settings

As previously mentioned, definitions of hypo- and hyperfibrinolysis are less clear-cut than the definitions of hypo- and hypercoagulable states. That being said, disturbed fibrinolysis is recognised as a part of the pathophysiology in a wide range of clinical conditions and has been correlated with clinical outcomes. Inherited disorders of fibrinolysis are rare, while acquired hyper- or hypofibrinolysis is more common. Acquired hyperfibrinolysis is separated into primary or secondary based on whether fibrinolysis is primarily increased by the presence of tPA (primary) or the absence of fibrinolysis inhibitors (secondary) [63]: primary hyperfibrinolysis can be seen in, e.g., acute promyelocytic leukaemia, orthotopic liver transplantation and post-partum haemorrhage, while secondary hyperfibrinolysis can follow excessive activation of coagulation, e.g., cardio-pulmonary bypass, or in chronic liver disease. Trauma is classified as both primary and secondary hyperfibrinolysis.

Acquired hypofibrinolysis is often related to substantially elevated PAI-1 levels. Elevated PAI-1 levels are found in obesity [64], aging, thrombosis, trauma, COVID-19 [65], sepsis [66] and in postoperative fibrinolytic shutdown [67]. Furthermore, changes in the amount of α2-antiplasmin incorporated in the clot structure and levels of FXIII [68], as well as increased procoagulant activity and fibrinogen concentrations, may change susceptibility to lysis [69].

The following sections provides an overview of our current knowledge of changes in fibrinolysis in selected clinical conditions and of the potential therapeutic implications (Table 1).

### 4.1. Fibrinolysis in Trauma

Trauma is a leading cause of death worldwide, particularly among the young. One-quarter to one-third of trauma patients suffer from traumatic coagulopathy. Kashuk et al. introduced the concept of the “Bloody vicious cycle” in the early 1980s where coagulopathy, hypothermia, acidosis, and tissue hypoxia pushes the haemostatic balance towards bleeding, which again aggravates hypoperfusion, hypoxia, and consumptive coagulopathy and speeds up the vicious cycle [70]. This was increasingly recognised as leading to higher mortality rates and led to the formalisation of damage control surgery and resuscitation techniques [71]. In the 2000s, Brohi, Cohen, and colleagues described the entity of trauma-induced coagulopathy having a significant impact on functional outcome and mortality [71]. Trauma-induced coagulopathy is considered a multifactorial phenomenon, with changes in both clot formation and fibrinolysis, in combination with the failure of vascular homeostasis and immunoactivation [72]. The following will focus on changes in fibrinolysis in major trauma and brain trauma.

#### 4.1.1. Major Trauma

The fibrinolytic system is activated in severe trauma with three recognised phenotypes of fibrinolysis in trauma: hyperfibrinolysis, hypofibrinolysis, and fibrinolytic shutdown [73]. Elevated PAP levels on admission to hospital is seen in 90% of trauma patients and indicates prior activation of fibrinolysis and subsequent shutdown [74]. Ongoing hyperfibrinolysis at admission measured by viscoelastic assays is present in less than 20% of the most severely injured trauma patients [71], but is associated with a high mortality rate and massive transfusion requirements.

Low fibrinolytic activity in trauma is also associated with increased mortality. Currently two definitions dominate, hypofibrinolysis and fibrinolytic shutdown, with fibrinolytic shutdown being the most prevalent. Hypofibrinolysis is a condition with impaired activation of fibrinolysis per se, and fibrinolytic shutdown is characterised by initial activation of fibrinolysis with subsequent shutdown by substantial release of PAI-1 [73]. Patients with fibrinolytic shutdown often have prolonged prothrombin time, platelet dysfunction, and low fibrinogen levels, and do not generally benefit from antifibrinolytic treatment, though it may be beneficial in selected patients with impaired platelet function and/or prolonged clotting time [75]. Furthermore, fibrinolytic shutdown beyond 24 h of injury is associated with increased mortality and ventilator requirements. The Clinical Randomisation of an Antifibrinolytic in Significant Hemorrhage 2 (CRASH-2) trial evaluating the effect of tranexamic acid versus placebo in severe trauma with suspected or present haemorrhagic shock has led to a widespread use of tranexamic acid in the trauma setting. It must be held in mind that tranexamic acid administrated later than 3 h after injury increased mortality [5], and recent studies have not confirmed the findings of the CRASH-studies [76,77,78].

#### 4.1.2. Brain Trauma

Alterations in markers of fibrinolysis in traumatic brain injury (TBI) may be useful for predicting neurological outcome, but may also guide clinical management. A TEG^®^-guided classification of fibrinolytic phenotypes, hyperfibrinolysis, physiologic fibrinolysis and fibrinolytic shutdown, has been suggested [79,80], but rests on observed mortality rates rather than specific changes in fibrinolytic markers [81]. However, the elevation of D-dimer levels has repeatedly been associated with poor outcome in TBI, as have low levels of fibrinogen [80,81], and dysregulated fibrinolysis adds to the risk of progressive intracerebral haemorrhage following TBI [81]. Investigating the time course of changes in coagulation and fibrinolysis in 61 TBI patients, Nakae et al. [82] reported that patients with a poor outcome had significantly higher levels of PAI-1 and D-dimer on the first day than patients with a good outcome. A positive correlation was found between D-dimer and PAI-1, and these findings were interpreted as hypercoagulability on admission was followed by increased fibrinolysis (D-dimer) and subsequent fibrinolytic shutdown (increased PAI-1). To some degree, these results conflict with Samuels et al. reporting hypocoagulation, but also some degree of fibrinolytic shutdown in TBI patients, when compared to trauma patients with no TBI [80]. The CRASH-3 trial, in which almost 13,000 patients with isolated TBI were randomised to tranexamic acid or placebo, the administration of tranexamic acid within 3 h of injury provided significant clinical benefit, but only in patients with mild-to-moderate brain injury, and had no effect in patients with more severe brain injury [83]. The results from the CRASH-3 trial have by others been interpreted as tranexamic acid should only be administrated to TBI patients with a Glasgow come scale score of 9–12 [84]. The diversity of these results and conclusions emphasise that we need better fibrinolytic markers to classify fibrinolytic phenotype.

### 4.2. Fibrinolysis in Coronary Artery Disease

Despite improved treatment options, coronary artery disease (CAD) is still the number one cause of mortality in the Western world, and the prevalence is increasing worldwide. Established risk factors such as smoking, hypercholesterolemia, and hypertension can be targeted through life style changes or pharmacological prophylaxis, but only partly explain the pathophysiology of CAD. Regarding acute coronary syndrome (ACS), thrombolytic therapy has been used since the late 1950s for the treatment of acute myocardial infarction (AMI) to dissolve the coronary thrombus [85]. However, thrombolysis is now only recommended in ST-elevation myocardial infarction if primary percutaneous coronary intervention (PCI) is not available within 120 min of diagnosis [86]. Changes in fibrinolytic capacity may be present already during early stages of CAD and may contribute to disease progression. Studies consistently found decreased clot permeability and prolonged lysis times in patients with stable CAD [87,88,89], as well as in ACS patients [90,91]. Traditional CAD risk factors may partly explain this, [92] especially diabetes mellitus. PAI-1 release from the atherosclerotic endothelium may be an important contributor to hypofibrinolysis in CAD. Increased local thrombin formation around the atherosclerotic plaque may also contribute to decreased clot permeability. Furthermore, sustained hypofibrinolysis is associated with unfavourable outcomes in CAD patients, including the development of ACS in stable CAD [93] and ACS recurrence and cardiovascular mortality in ACS patients [94,95]. Though the role of fibrinolysis in CAD is increasingly recognised, the clinical consequence is yet limited, since no profibrinolytic agents are approved for treatment of global, low-grade hypofibrinolysis.

### 4.3. Fibrinolysis in Sepsis

The definition of sepsis has changed significantly over the years, but is currently defined as “a life-threatening organ dysfunction caused by a dysregulated host response to infection”. Septic shock is defined as sepsis with need for vasopressor treatment to maintain mean arterial pressure > 65 mmHg or serum lactate > 2 mmol/L in the absence of hypovolemia [96]. Sepsis presents with varying degrees of organ dysfunction, evaluated by use of the Sequential Organ Failure Assessments (SOFA) score, which correlates with mortality [97,98]. Regarding changes in coagulation, focus has been on the activation of platelets and thrombin in interaction with the immune system [99]. However, in recent years the presence and importance of hypofibrinolysis in sepsis has come into play [66].

A consistent finding in sepsis is hypofibrinolysis evaluated by both viscoelastic assays and dynamic plasma-based assays [52,100,101]. These findings are accompanied by increased PAI-1, decreased plasminogen and increased fibrinogen. The increased fibrinogen may lead to a denser and more lysis resistant clot structure. Endothelial activation leads to high levels of circulating tPA, followed by a more sustained increase in PAI-1. In later stages of sepsis, plasminogen consumption and decreased synthesis will lead to lower levels of circulating plasminogen [66]. Furthermore, extensive TAFI activation is seen early in sepsis [102]. Hypofibrinolysis in sepsis may be associated with increase in organ failure and mortality, but consistent associations with outcome remain to be clarified [66].

Since suPAR is shed during inflammation, it has naturally been investigated as a prognostic marker in sepsis. Two recent meta-analyses showed moderate-to-good ability for suPAR to predict mortality with area under receiver operator characteristics (ROC) curves of approximately 0.65–0.80 [103,104]. It is yet unknown whether suPAR contributes to altered fibrinolytic capacity in sepsis through interactions with uPA.

### 4.4. Venous Thromboembolism

Venous thromboembolism (VTE) covers the spectrum from subclinical deep venous thrombosis (DVT) to sudden death by pulmonary embolism (PE). PE and DVT are often seen concurrently. Hence, the two conditions and the biochemical changes related to them cannot be completely separated. However, in acute VTE, clot lysis time may be shorter in PE than in DVT without PE, possibly indicating a looser fibrin network, predisposing to clot fragmentation and the development of PE from DVT [69].

In a population-based nested case-control study from Tromsø, Norway, the future risk of VTE increased in a dose-dependent manner with increasing PAI-1 at baseline, and PAI-1 explained approximately 15% of the VTE risk in obesity [64]. Following documented VTE, impaired fibrin clot lysis is seen, and patients later developing recurrent VTE have higher PAI-1 levels than patients not experiencing recurrent VTE [69]. As much as 77% of variation in clot lysis time in DVT patients is attributed to PAI-1, TAFI, and α2-antiplasmin. Furthermore, among patients with long clot lysis times, oral contraceptives, immobilisation, and the presence of factor V Leiden mutation increases the risk of VTE substantially in comparison to patients with short clot lysis times [69]. This indicates synergistic effects of alterations in the coagulation and fibrinolytic systems on VTE risk.

#### Pulmonary Embolism

Patients with PE may have looser clot structure compared to patients with isolated DVT. However, intermediate-to-high-risk PE patients have significantly longer lysis time than low-and-intermediate-risk PE patients [69]. High PAI-1, low TAFI, and low α2-antiplasmin indicate a high-risk biomarker profile in acute PE [105]. This may be of clinical importance when evaluating the results of catheter-based therapy and thrombolysis in PE. PE complicated with hemodynamic instability (shock or hypotension) should be treated rapidly with reperfusion techniques, particularly systemic thrombolysis [106]. The greatest benefit is observed when treatment is initiated within 48 h, but thrombolysis can still be useful in patients who have had symptoms for 6–14 days. However, this treatment comes with a high risk of major bleeding, particularly intracerebral haemorrhage, and in normotensive patients with inter-mediate risk PE, the risk reduction in hemodynamic decompensation is outweighed by an increased risk of major bleeding [15]. Catheter-based low-dose thrombolytic therapy can limit the dose of thrombolysis by a factor of four. Furthermore, the delivery of high-frequency ultrasound within the thrombus may enhance the action of the thrombolytic therapy, but results are contradictive and primarily evaluated by right ventricular function and not mortality or recurrent VTE [106]. Thrombo-aspiration may be a tempting alternative or supplemental to local thrombolysis and is recommended as an alternative when systemic thrombolysis is contraindicated. Thrombo-aspiration may improve right ventricular function, but there is a lack of studies evaluating gain with regards to mortality or recurrence of VTE.

### 4.5. Fibrinolysis in Liver Disease

Patients with liver disease have been described to have accelerated fibrinolysis [107], and it is generally accepted that this hyperfibrinolytic state contributes to bleeding [108]. However, when evaluating different aspects of liver disease, nuances can be added.

#### 4.5.1. Cirrhosis

In stable cirrhosis, fibrinolysis is possibly increased [109,110,111,112] as tPA is released from endothelium, and clearance of tPA is reduced due to the diseased liver. However, the fibrinolytic activity may depend on the aetiology of cirrhosis. In a study comparing patients with alcoholic cirrhosis (*n* = 15) and non-alcoholic steatohepatitis (NASH) cirrhosis (*n* = 22), patients with NASH cirrhosis had significantly prolonged clot lysis time, indicating a hypofibrinolytic state, in comparison to patients with alcoholic cirrhosis and healthy controls [113]. This is in accordance with others who also described hypofibrinolysis in patients with NASH [114,115] and cholestatic disease [115]. Plasma levels of PAI-1 were higher in these two patient groups compared with other aetiologies of cirrhosis and healthy controls. This hypofibrinolytic state in NASH may explain the increased risk of thrombotic events observed in these patients [116]. In both alcoholic and non-alcoholic cirrhosis, hepatocellular carcinoma is associated with hypofibrinolysis [117]. A clear association between hyperfibrinolysis and bleeding in stable liver cirrhosis has not been described, but cirrhosis-related bleeding is often not related to haemostatic failure but manifests primarily as variceal bleeding and bleeding after invasive procedures, where portal hypertension and local vascular abnormalities are major aetiological factors of the bleed. Many post-procedural bleedings are likely related to inadvertent vessel wall injury [118,119].

In patients with acutely decompensated cirrhosis or acute-on-chronic liver failure, test results are highly variable [108,120]. Acutely decompensated cirrhosis patients generally show hyperfibrinolytic traits in comparison to healthy controls [108], and hyperfibrinolysis may relate to the presence and degree of ascites [121]. Patients with acute-on-chronic liver failure show great variation ranging from severely hypofibrinolytic to hyperfibrinolytic when evaluated by dynamic clot lysis assays [120]. However, clot lysis times were not associated with bleeding in these patients [120].

#### 4.5.2. Acute Liver Failure and Liver Transplantation

Acute liver failure patients are for the most part in a profound hypofibrinolytic state [122]. However, this is not related to short-term mortality, and the importance is uncertain [123]. Whether there is a role for hypofibrinolysis in disease progression in acute liver failure remains to be elucidated.

In contrast, during orthotopic liver transplantation hyperfibrinolysis is prevalent. The defective clearance of tPA during the anhepatic phase and increased release of tPA from injured endothelium in the donor liver after the anhepatic phase, when oxygen-rich blood is reintroduced after a period of ischemia, may be the cause. In this situation, hyperfibrinolysis is associated with increased perioperative blood loss, as demonstrated by increased transfusion requirements [124,125]. A randomised trial administering prophylactic antifibrinolytic therapy reduced transfusions [126] and the use of aprotinine was not associated with thrombotic complications [127]. On this basis, antifibrinolytic therapy is recommended to treat or prevent hyperfibrinolysis in patients undergoing liver transplantation. The evaluation of haemostasis and fibrinolysis during surgery may be aided by the use of ecarin containing viscoelastometric testing [128].

### 4.6. Beyond Bleeding and Thrombosis Risk: Hereditary Angioedema

Hereditary angioedema is a condition characterised by episodes of increased vascular permeability which leads to swelling in the deep layers of skin and mucosa/submucosa and, depending on the localisation (e.g., airways), can be life-threatening [129]. The increase in vascular permeability is caused by the vasoactive substance bradykinin, an activation product of the contact system. The background for increased bradykinin production in hereditary angioedema is most often due to a lack of C1-inhibitor, a regulator of contact system activation [129]. However, one form of angioedema has been associated with a mutation in the plasminogen gene which renders plasminogen more susceptible to activation by tPA and uPA [130]. Under physiological circumstances, there is significant crosstalk between the contact and fibrinolytic systems, as they can activate each other in turn, and this crosstalk is probably enhanced in angioedema patients [30]. In hereditary angioedema, there is evidence of increased intra- and extravascular plasmin activation during swelling episodes, indicated by increased circulating PAP complex [131,132], decreasing PAI-1 [133] and the upregulation of uPAR in leukocytes [134]. This may in turn aggravate contact and immune activation. Thus, it is tempting to consider the fibrinolytic system as a treatment target in hereditary angioedema. However, while antifibrinolytic agents have been found to be beneficial in early studies, they are probably not as effective as newer drugs targeting the contact system which include recombinant C1-inhibitor and blockers of bradykinin and kallikrein [135].

## 5. The Fibrinolytic System as a Treatment Target

Anti-fibrinolytic therapy has gone from strength to strength since the presentation of the CRASH trials, and its use has increased substantially. Its price and availability makes it an attractive choice in the prehospital and trauma setting, as well as in low-income countries. However, the studies of trauma patients indicate that time is of the essence, as mortality increases if tranexamic acid is administered after a certain time window. Furthermore, fibrinolytic properties change with the injury severity, degree of haemorrhagic shock, and at a particular time after trauma. As discussed by Honeybul et al., tranexamic acid may not be as innocuous as currently believed, and future studies need to focus on the timing and applicability of tranexamic acid administration [84].

While thrombolysis is now less of a cornerstone in the treatment of acute myocardial infarction, it certainly has a place in the treatment of pulmonary embolism and stroke. Currently, a phase-3 study is evaluating the effect of non-immunogenic recombinant staphylokinase versus alteplase in massive PE. Further studies focus on catheter-based low-dose local thrombolysis and thrombo-aspiration in both pulmonary embolism and stroke.

Regarding the treatment of global hypofibrinolysis in, e.g., sepsis- or trauma-associated fibrinolytic shutdown, there is currently no approved pharmacological therapy. The inhibition of antifibrinolytic proteins has been researched for several decades. Inhibitors of PAI-1 [136,137], α2-antiplasmin [138], and TAFI [139] have been shown to alleviate thrombus formation in several animal models, but have not so far progressed to clinical trials, partly due to increased bleeding risk. Furthermore, low-dose systemic tPA in lysis-resistant critical illness may hold potential if reliable biochemical assays allow us to identify which patients will benefit from this treatment [140].

## 6. Conclusions and Future Directions

In the last 75 years, the map of the fibrinolytic system has been established and is still expanding. The routine coagulation laboratory assays still do not allow us to discriminate between hyper- and hypofibrinolysis in the diagnostic setting, but particularly viscoelastic assays hold potential in this matter. Studies applying viscoelastic assays, modified to identify changes in fibrinolysis related to clinical outcome are awaited, as this will identify patients who will benefit from antifibrinolytic or profibrinolytic treatment. Particularly in major trauma and brain trauma, fibrinolysis changes with time and may shift from hyper- to hypofibrinolysis, closing the window for antifibrinolytic treatment within a short time after trauma.

In sepsis and VTE, hypofibrinolysis predominates, but treatment options are limited and awaited. Inhibitors of PAI-1, α2-antiplasmin, and TAFI may in time progress to clinical trials, as may low-dose thrombolysis. In pulmonary embolisms, the refinement of thrombolysis and mechanical removal of the clot are part of ongoing clinical trials. In conclusion, investigations of fibrinolysis in the clinical setting, as well as the further development of laboratory assays, are needed, and hold potential for major improvements in patient care and outcome.

## Figures and Tables

**Figure 1 ijms-24-14179-f001:**
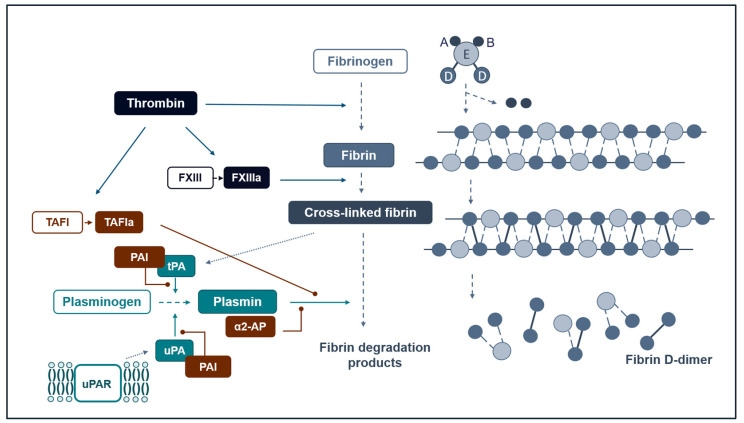
Schematic overview of activators and regulators of fibrinolysis. As a result of coagulation activation, fibrinogen A and B domains are cleaved from fibrinogen by thrombin, giving rise to fibrin monomers, which subsequently polymerise to form the fibrin clot. The clot is then stabilised by covalent crosslinking of D-domains induced by coagulation factor (F) XIIIa. Plasmin is the main enzyme responsible for fibrin clot degradation (fibrinolysis) and is activated by tissue-type plasminogen activator (tPA) in the presence of fibrin, or by urokinase-type plasminogen activator (uPA) in the presence of its cellular receptor (uPAR). Important regulators of fibrinolysis are plasminogen activator inhibitors (PAI)-1 and -2, which inhibit tPA and uPA activity; α2-antiplasmin binds and inhibits plasmin directly, and thrombin-activatable fibrinolysis inhibitor (TAFI) cleaves lysine residues from the fibrin clot and thereby impedes plasminogen binding to fibrin. (a) denotes activated enzyme.

**Table 1 ijms-24-14179-t001:** Changes in the fibrinolytic system in specific clinical settings.

Clinical Setting	Changes in the Fibrinolytic System
** *Liver disease* **	
Stable cirrhosis	Alcoholic cirrhosis: increased fibrinolysis due to release and defective clearance of tPA Non-alcoholic steatohepatitis cirrhosis: hypofibrinolysis due to increased levels of PAI-1
Decompensated cirrhosis and acute-on-chronic liver failure	Highly variable, ranging from severely hypofibrinolytic to hyperfibrinolytic
Acute liver failure	Profound hypofibrinolysis, uncertain clinical relevance
Liver transplantation	Hyperfibrinolysis due to defective clearance of tPA in the anhepatic state and increased release from the donor liver. Antifibrinolytic therapy is recommended during surgery
** *Trauma* **	
Major trauma	Hyperfibrinolysis, hypofibrinolysis, and fibrinolytic shutdown. Antifibrinolytic therapy increases survival after major trauma with haemorrhagic shock if administered less than 3 h after trauma
Brain trauma	Variable, both hyperfibrinolysis and hypofibrinolysis. Early antifibrinolytic therapy may be beneficial in mild-to-moderate brain injury
Sepsis	Consistent findings of hypofibrinolysis in correlation with organ failure. Increased levels of PAI-1, TAFIa/TAFIa, and fibrinogen, decreased levels of plasminogen
Cardiovascular disease	Both stable atherosclerosis and ACS are associated with decreased fibrin clot permeability and prolonged lysis times. Clot structure and lysis time may be prognostic marker for reinfarction or cardiovascular mortality in ACS patients
** *Venous thromboembolism (VTE)* **	
DVT	A high PAI-1 level predisposes to VTE. Patients with recurrent DVT have higher PAI-levels than patients without recurrence of DVT.
PE	Patients with PE may have looser clot structure than patients with isolated DVT. In high-risk PE patients, systemic thrombolysis is indicated. In cases of contraindications, catheter-based thrombo-aspiration is an alternative.

Abbreviations: ACS: acute coronary syndrome; DVT: deep vein thrombosis; PAI-1: plasminogen activator inhibitor-1; PE: pulmonary embolism; TAFI: thrombin-activatable fibrinolysis inhibitor; tPA: tissue plasminogen activator; and VTE: venous thromboembolism.

## Data Availability

Not applicable.

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
