# Peer review of "The Fibrinolytic System and Its Measurement: History, Current Uses and Future Directions for Diagnosis and Treatment"

_ijms, 2023, doi:10.3390/ijms241814179_

Round 1

Reviewer 1 Report

The Review entitled "The fibrinolytic system and its measurement: History, current uses and future directions for diagnosis and treatment" is very interesting and easy to understand.

However, there are some elements that should be included in the manuscript.

1) It is important to describe the soluble form of uPAR (suPAR) as a marker of kidney disease and inflammatory disorders. In patients admitted to acute care, systemic levels of suPAR correlate positively with markers of organ dysfunction and severity of the disease.

2) Among fibrinolytc factors, suPAR has emerged as a promising sepsis marker (IJMS 2023, 24, 12376 doi: 10.3390/ijms241512376). To make paragraph 4.3 complete, the Authors should talk about it.

3) The role of the Plasminogen System in Angioedema has been extensively investigated (JCM 2021, 10(3):518. doi: 10.3390/jcm10030518). In this Review, the Authors should put this topic in the Section 4 entitled "Fibrinolysis in specific clinical settings".

Reviewer 2 Report

Content suggestions:

1. I would like to kindly ask the authors to provide the exact recommendations of fibrinolytic therapy (its indications, time interval needed to be kept to be meaningful...).

2. What are the complications of such therapy ? Please, describe them.

From my point of view, after the addition of the responses to my questions, the manuscript can be accepted for the publication.

Therefore, I recommend minor revision.
